# Novel Dihydrocoumarins Induced by Radiolysis as Potent Tyrosinase Inhibitors

**DOI:** 10.3390/molecules29020341

**Published:** 2024-01-10

**Authors:** Gyeong Han Jeong, Manisha Yadav, Seung Sik Lee, Byung Yeoup Chung, Jae-Hyeon Cho, In-Chul Lee, Hyoung-Woo Bai, Tae Hoon Kim

**Affiliations:** 1Research Division for Biotechnology, Advanced Radiation Technology Institute (ARTI), Korea Atomic Energy Research Institute (KAERI), Jeongeup 56212, Republic of Korea; jkh4598@kaeri.re.kr (G.H.J.); sslee@kaeri.re.kr (S.S.L.); bychung@kaeri.re.kr (B.Y.C.); 2Center for Companion Animal New Drug Development, Korea Institute of Toxicology (KIT), Jeongeup 56212, Republic of Korea; 3Department of Food Science and Biotechnology, Daegu University, Gyeongsan 38453, Republic of Korea; manishayv41@gmail.com; 4Radiation Biotechnology and Applied Radioisotope Science, University of Science and Technology (UST), Daejeon 34113, Republic of Korea; 5Institute of Animal Medicine, College of Veterinary Medicine, Gyeongsang National University, Jinju 52828, Republic of Korea; jaehcho@gun.ac.kr; 6Department of Cosmetic Science and Technology, Seowon University, Cheongju 28674, Republic of Korea; lic9418@seowon.ac.kr

**Keywords:** 4-methylumbelliferone, gamma irradiation, tyrosinase inhibition, hydrogenation, hydroxymethylation

## Abstract

A representative naturally occurring coumarin, 4-methylumbelliferone (**5**), was exposed to 50 kGy of gamma ray, resulting in four newly generated dihydrocoumarin products **1**–**4** induced by the gamma irradiation. The structures of these new products were elucidated by interpretation of spectroscopic data (NMR, MS, [α]_D_, and UV). The unusual bisdihydrocoumarin **4** exhibited improved tyrosinase inhibitory capacity toward mushroom tyrosinase with IC_50_ values of 19.8 ± 0.5 μM as compared to the original 4-methylumbelliferone (**5**). A kinetic analysis also exhibited that the potent metabolite **4** had non-competitive modes of action. Linkage of the hydroxymethyl group in the C-3 and C-4 positions on the lactone ring probably enhances the tyrosinase inhibitory effect of 4-methylumbelliferone (**5**). Thus, the novel coumarin analog **4** is an interesting new class of tyrosinase inhibitory candidates that requires further examination.

## 1. Introduction

Tyrosinase is a well-known key enzyme of melanin biosynthesis in microorganisms, plants, and animals. Specifically, it plays an important role in the hydroxylation of l-tyrosine to l-3,4-dihydroxyphenylalanine (l-DOPA), and in the oxidation of l-DOPA to dopaquinone [1]. The resulting dopaquinone is further converted into dopachrome 5,6-dihydroxyindole and indole-5,6-quinone, which then undergoes cyclization and subsequent polymerization to form melanin. Additionally, tyrosinase functions as both a tyrosine hydroxylase and a DOPA oxidase within the melanosome. As a tyrosine hydroxylase, it catalyzes the oxidation of tyrosine to DOPA. Subsequently, as a DOPA oxidase, it oxidizes DOPA to dopaquinone, playing a crucial role in melanin polymer synthesis [1]. Thus, this enzyme is responsible for the pigmentation of the skin, eyes, and hair [2]. Tyrosinase inhibitors are mainly used as depigmenting agents in cosmetics and pharmaceuticals for preventing and treating hyperpigmentation disorders in the epidermal layers of the human skin [3]. Most of the anti-tyrosinase agents are derived from natural resources, including several plants and terrestrial or marine microorganisms [4]. Among the tyrosinase inhibitors, azelaic acid, kojic acid, and ellagic acid are the most representative natural whitening agents [5]. Despite the clinical and industrial increase in demands for effective tyrosinase inhibitors, currently, there are very few candidates certified for clinical use [6], resulting in a strong requirement to discover potent tyrosinase inhibitory compounds.

Coumarins are well-known secondary metabolites widely distributed in diverse plants and are abundant in the leaves, bark, and roots of *Rutaceae* and *Umbelliferae* plants. The major biosynthetic pathway of coumarin occurs via shikimic acid and cinnamic acid, through phenylalanine metabolism [7]. This type of metabolite is structurally represented by benzo-α-pyrone (2*H*-1-benzopiran-2-one). Natural coumarins are subdivided into different sub-classes comprising coumarins, isocoumarins, furanocoumarins, pyranocoumairns, biscoumarins, and phenylcoumarins [8]. To date, more than 1300 types of coumarin have been identified as natural and synthesized compounds. Recent reports indicate that coumarins possess several attractive characteristics such as simple chemical structure, low molecular weight, high bioavailability, high solubility, and low toxicity, ensuring their prominent role as lead compounds in the drug discovery industry [9]. Several pharmacological properties attributed to coumarins include antioxidative, antimicrobial, anti-inflammatory, anticoagulant, antidiabetic, and neuroprotective effects [10]. These compounds have therefore attracted a huge pharmacological interest over the last decade. This advantageous skeleton provides a valuable platform for designing chemical libraries while exploring new drug candidates.

Gamma irradiation is an established advanced strategy applied in food processing. It is also recognized to play a major role in the destruction of microorganisms due to the abundance of reactive species and free radicals such as methoxy (CH_3_O^•^), hydroxy alkyl (^•^CH_2_OH), hydrogen (H^•^), superoxide anion (O_2_^•−^), peroxyl (OOH^•^) radicals, as well as hydroxyl ion (OH^−^), molecular hydrogen (H_2_), and H_2_O [10]. However, very few studies have researched the influence of gamma radiolysis on the phenolic constituents widely distributed in natural sources. The degradation reaction of representative coumarin and the formation of other compounds via gamma-ray treatment are still uncertain. The chemical change in naturally occurring polyphenolics induced by gamma radiolysis is closely associated with human health. Therefore, accurately estimating their levels under gamma-ray treatment conditions is crucial for assessing their potential biological properties and ensuring their safety [11,12]. In a previous study, we irradiated rotenone with gamma rays and confirmed that the new derivatives, rotenoisins A and B, were formed and inhibited preadipocyte differentiation in 3T3-L1 cells [13]. Additionally, we reported that irradiation of the natural flavonoid, baicalin, produced hydroxymethylated products with potent anti-inflammatory effects [12]. Our recent study suggests that gamma irradiation is a unique chemical procedure for the creation of structurally novel, biologically active compounds with enhanced convenience and yields [12,13,14]. This study evaluates the changes in the chemical structures and biological capacities of 4-methylumbelliferone induced by gamma irradiation.

## 2. Results and Discussion

### 2.1. Isolation and Structure Elucidation of Newly Generated Dihydrocoumarins

A methanolic sample solution containing pure 4-methylumbelliferone (4-MUF) was transferred to a container and immediately irradiated using cobalt-60 gamma rays; the transferred products were continuously monitored using reversed-phase HPLC. The evaporated irradiated sample (50 kGy) showed significantly improved inhibitory capacity with an IC_50_ value of 86.7 ± 1.6 μg/mL against mushroom tyrosinase. Careful column chromatographic isolation led to the purification of four new dihydrocoumarin derivatives: radiocoumarones A (**1**), B (**2**), C (**3**), and D (**4**) (Figure 1). The rare hydroxymethylated dihyrocoumarin analogs (**1**–**4**) were verified to contain rare functional groups in the C-3 or C-4 positions on the lactone ring of 4-methylumbelliferone.

Compound **1** was purified as a white amorphous powder with a molecular formula of C_11_H_12_O_4_, which was determined using a combination of the ^13^C NMR spectroscopic data and the positive-mode sodiated HRESIMS at *m*/*z* 231.0626 [M + Na]^+^ (calcd. for C_11_H_12_O_4_Na). The UV spectrum displayed characteristic absorption peak maxima at 215 (log ε 2.27) and 275 (log ε 0.61) [15]. The ^1^H NMR spectrum of **1** in CD_3_OD exhibited characteristic ABX-type aromatic signals at *δ*_H_ 6.28 (1H, dd, *J* = 8.4, 2.4 Hz, H-6), 6.31 (1H, d, *J* = 2.4 Hz, H-8), and 6.86 (1H, d, *J* = 8.4 Hz, H-5), indicating the occurrence of a 1,3,4-trisubstituted benzene ring in **1**. The spectrum also showed a methylene proton at *δ*_H_ 2.63 (1H, d, *J* = 16.8 Hz, H-3β), 3.05 (1H, d, *J* = 16.8 Hz, H-3α), a methyl signal at *δ*_H_ 1.45 (3H, s, H-12), and diagnostic hydroxymethyl protons at *δ*_H_ 4.47 (1H, d, *J* = 9.0 Hz, H-11), 4.59 (1H, d, *J* = 9.0 Hz, H-11), indicating the characteristic resonances in the dihydrocoumarin [12] framework linked by one hydroxymethyl group. Consistent with these ^1^H NMR elucidations, this structure was supported by the appearance of ^13^C NMR resonances (Table 1) and HSQC interpretations. The ^13^C NMR and HSQC experiments of **1** further displayed the occurrence of signals for trisubstituted aromatic rings at *δ*_C_ 102.7 (C-8), 106.1 (C-6), 121.5 (C-10), 127.0 (C-5), 155.9 (C-9), 157.2 (C-7), one characteristic hydroxymethyl carbon at *δ*_C_ 79.1 (C-11), one methyl group at *δ*_C_ 24.8 (C-12), one methylene carbon at *δ*_C_ 41.5 (C-3), one carbonyl carbon at *δ*_C_ 178.6 (C-2), and characteristic quaternary carbon at *δ*_C_ 42.5 (C-4) (Table 1). These resonances closely resemble the parent 4-methylumbelliferone [16], except for the appearance of a hydroxymethyl and a methylene group instead of the olefine moiety at the C-3 and C-4 in **1**. The HMBC cross-peaks of H-11 and H-12 to C-4 corroborated the locations of the methyl and hydroxymethyl functionalities at the C-4 of the lactone ring (Figure 2A). The relative stereostructure of the C-4 chiral center in **1** was characterized by the NOESY correlations between H-3β/H-12 (CH_3_) and H-3α/H-11 (Figure 2B). On the basis of these findings, the planner structure **1** was assigned as radiocoumarone A, which is a new dihyrocoumarin induced by gamma radiolysis.

The HRESIMS of compound **2** gave molecular ion peaks at *m*/*z* 261.0732 [M + Na]^+^ (calcd. for C_12_H_14_O_5_Na, 261.0733) with a molecular formula of C_12_H_14_O_5_, which contains one more hydroxymethyl residue than that of **1**. The ^1^H and ^13^C NMR spectral data of **2** were closely comparable to those of dihydrocoumarin **1**, and the only difference identified was the presence of a hydroxymethyl signal at H-3 instead of the H-3 proton signal observed in **1**. The connective positions of one methyl and two hydroxymethyl units in **2** were further supported by the key HMBC relationships from H-3β (*δ*_H_ 2.91)/H-11 (*δ*_H_ 3.79 and 3.66) to C-2 (*δ*_C_ 180.9)/C-3 (*δ*_C_ 53.6) and from H-12 (*δ*_H_ 4.84 and 4.32)/H-13 (*δ*_H_ 1.45) to C-4 (*δ*_C_ 44.3) (Figure 2A). The relative stereochemistry of the C-3 and C-4 positions in the lactone ring were characterized by the NOESY spectrum (Figure 2B). The spectrum of **2** displayed correlations between H-3β/H-13 (CH_3_) and H-11/H-12, indicating 3*S**, 4*R** configuration between hydroxymethyl (H-11) and methyl (H-13) groups. Based on the above evidence, the structure of the new compound **2** was assigned as radiocoumarone B, which is a new radiolytic transformation product of 4-methylumbelliferone.

Compound **3** was obtained as a white amorphous powder. The HRESIMS displayed a pseudomolecular ion peak at *m*/*z* 291.0831 [M + Na]^+^, corresponding to the molecular formula C_13_H_16_O_6_. In the ^1^H NMR spectrum of **3** (Table 1), ABX-type aromatic signals at *δ*_H_ 6.30 (1H, dd, *J* = 8.4, 2.4 Hz, H-6), 6.34 (1H, d, *J* = 2.4 Hz, H-8), and 7.08 (1H, d, *J* = 8.4 Hz, H-5) were recognized as belonging to the aromatic moiety of the dihydrocoumarin skeleton. The ^1^H NMR spectrum of **3** also exhibited resonances corresponding to two hydroxymethylene groups at *δ*_H_ 4.06 (1H, dd, *J* = 11.4, 4.2 Hz, H-11), 4.13 (1H, dd, *J* = 11.4, 4.2 Hz, H-11), 4.61 (1H, d, *J* = 9.0 Hz, H-12), and 4.80 (1H, d, *J* = 9.0 Hz, H-12), and one hydroxyethylene group at *δ*_H_ 2.10 (1H, m, H-13), 2.63 (1H, m, H-13), 3.29 (1H, m, H-14), and 3.42 (1H, m, H-14). The ^1^H and ^13^C NMR spectral analysis of **3** was similar to **2**, and the only difference identified in **3** was the presence of a hydroxyethyl moiety at C-4 instead of a methyl group. The further partial structure, CH_2_-CH_2_-OH, was inferred from the HMBC relationships of H-13 to C-4/C-14 and H-14 to C-4/C-13. In addition, the HMBC correlations of the two hydroxymethyl groups at H-11 and H-12 to C-3 and C-4 supported the location of the hydroxymethyl groups at C-3 and C-4, respectively (Figure 2A). The relative configurations of the chiral centers at C-3 and C-4 in **3** were characterized by the NOESY correlations to be between H-3β/H-13 and H-11/H-12 (Figure 2B). Based on the above evidence, the structure of new **3** was assigned the trivial name radiocoumarone C.

Compound **4** was purified as a new product in the form of a white amorphous powder. A pseudomolecular ion peak at *m*/*z* 415.1381 [M + H]^+^ was observed in the positive HRESIMS in combination with ^13^C NMR spectroscopic data, consistent with the molecular formula C_22_H_22_O_8_. The simple absorption maxima with a broad band at 278 nm in the UV spectrum were ascribed to the presence of the dihydrocoumarin skeleton [12]. The ^1^H and ^13^C NMR spectra of **4** were also nearly similar to those of **2**, except for the evidence of one less hydroxymethyl proton at the C-4, suggesting that the *C*-*C* linkage is located at the C-4 in both monomeric halves of the dimer (Table 2). The linkage position of each dihydrocoumarin unit was established using the HMBC technique, which demonstrated a three-bond correlation between the methine proton signal (H-3) to C-4, -5, -12 (CH_3_), and -10 (Figure 2A). The relative stereochemistry of the chiral centers on the lactone ring was inferred by the key NOESY correlations between H-3 and H-12 (CH_3_). Thus, the new product of compound **4** was assigned the name radiocoumarone D. However, the absolute configurations of hydroxymethylated dihydrocoumarin **1**–**4** could not be determined due to the apparent unavailability of a relevant reference CD or optical rotation value in the literature.

In recent investigations, free radicals formed via methanolic radiolysis have been revealed to impart hydroxyalkyl functionality in some natural products such as genistein and luteolin [17,18]. In the present study, reactive molecular species and free radicals produced via gamma irradiation under methanolic conditions were capable of hydrogenation, hydroxymethylation, and dimerization of 4-methylumbelliferone, resulting in the formation of dihydrocoumarin products, viz., radiocoumarones A (**1**), B (**2**), C (**3**), and D (**4**).

### 2.2. Inhibition Effects of Mushroom Tyrosinase

Tyrosinase is a metalloenzyme that plays a primary role in the production of melanin from tyrosine when the skin is exposed to UV rays. Tyrosinase-related protein-1 (TRP-1) and dopachrome tautomerase (TRP-2) result in the formation of melanin as eumelanin (which appears black and brown) and pheomelanin (which appears yellow and red). Tyrosinase is present in epidermal melanocytes as well as in the pigment epithelia of the retina, iris, and ciliary body of the eye [19], and is one of the chief enzymes responsible for skin pigmentation in mammals. Melanin production can be inhibited by suppressing tyrosinase activity, thereby preventing the induction of melasma, freckles, and senile erythema [20]. Despite the many pharmacological capacities of coumarin, the skin protective property of coumarin has not yet been studied using gamma irradiation for tyrosinase inhibitory potential. Radiocatalytic hydrogenated molecules **1**–**4** obtained in the present study were evaluated for their inhibitory capacities toward mushroom tyrosinase [21]. Among the hydrogenated coumarins, radiocoumarones D (**4**) and B (**2**) displayed increased inhibitory activities compared to the parent 4-methylumbelliferone, with IC_50_ values of 19.8 ± 0.5 and 49.0 ± 1.3 μM, respectively (Table 3). After that, the inhibition kinetics of bisdihydrocoumarin **4** against tyrosinase-mediated l-DOPA oxidation was performed. As shown in Figure 3, the Lineweaver–Burk plots clearly demonstrated that radiocoumarone D functioned as a non-competitive mode of action. The *Ki* value of compound **4** was also determined as 16.0 ± 0.2 μM (Table 3). This result suggested that dimeric dihyrocoumarin **4,** having a *C*-*C* linkage at the C-4 position on the lactone ring, was found to display relatively higher inhibitory capacity than monomeric dihydrocoumarin **1**.

3,4-Dihydrocoumarin and coumarin are attractive candidates for further studies due to the widespread application of coumarin in cosmetics, perfumes, and other industries. Dihydrocoumarin is a minor coumarin metabolite in naturally occurring phytochemicals which, unlike coumarin, does not cause liver damage after chronic administration to rats [22]. Thus, structural dihydrogenation and additional modifications of representative coumarin may be valuable for improving bioavailability and bioefficacy. Current advances have recommended that reactive oxygen species and free radicals generated by gamma irradiation might be conveniently adapted to new molecules with higher biological properties [13]. The present study undertook the structural modification and isolation of structurally novel dihydrocoumarins containing hydroxy alkyl groups and validated the correlation with improved tyrosinase inhibitory capacities.

## 3. Materials and Methods

### 3.1. General Experimental Procedures

4-Methylumbelliferone and kojic acid were purchased from Sigma-Aldrich Corp. (St. Louis, MO, USA). All other chemical reagents purchased in this study were of analytical grade. UV spectra were measured on a Hitachi U-2000 spectrophotometer (Hitachi, Tokyo, Japan). ^1^H and ^13^C NMR spectra were performed on an Avance NEO-600 instrument (Bruker, Karlsruhe, Germany) operated at 600 and 150 MHz, respectively. Chemical shifts are given in *δ* (ppm) values comparative to those of CD_3_OD (*δ*_H_ 3.35; *δ*_C_ 49.0) and acetone-*d*_6_ (*δ*_H_ 2.04; *δ*_C_ 29.8) on a tetramethylsilane (TMS) scale. The standard pulse sequences programmed into the instruments were measured for each 2D operation. The *J*_CH_ value was set at 8 Hz in the HMBC spectra. ESI mass spectra were conducted on an Agilent HPLC-MS (Agilent Technologies, Palo Alto, CA, USA). Optical rotation was recorded using a P-2000 polarimeter (JASCO, Tokyo, Japan). Column chromatography was performed using Toyopearl HW-40 coarse grade (Tosoh Co., Tokyo, Japan) and YMC GEL ODS AQ 120-50S (YMC Co., Kyoto, Japan).

### 3.2. Sample Preparation and New Compound Isolation

Gamma irradiation of a sample solution comprising 4-methylumbelliferone (1.0 g) in methanol (1.0 L) in capped bottles was performed using a cobalt-60 γ source with an action of approximately 215 kCi and a dose rate of 50 kGy/h (absorbed dose). The irradiated 4-methylumbelliferone was dried immediately and successively monitored using reversed-phase HPLC. Thereafter, 500 mg of the irradiated sample was subjected to column chromatography over a Toyopearl HW-40 column (coarse grade; 2.8 cm i.d. × 35 cm) and a YMC GEL ODS AQ 120-50S column (1.1 cm i.d. × 41 cm) with water containing increasing amounts of MeOH in a stepwise gradient. This resulted in the isolation of pure new compounds **1** (33.5 mg, *t*_R_ 12.2 min), **2** (100.2 mg, *t*_R_ 10.5 min), **3** (7.6 mg, *t*_R_ 6.3 min), and **4** (21.3 mg, *t*_R_ 8.4 min). HPLC analysis was conducted on a YMC-Pack ODS A-302 column (4.6 mm i.d. × 150 mm; YMC Co., Ltd.); the solvent system comprised a linear gradient commencing with MeCN in 0.1% HCOOH/H_2_O (detection: UV 280 nm; flow rate: 1.0 mL/min; at 40 °C), increasing to 30% MeCN over 15 min, and then maximizing to 70% MeCN over 25 min (Appendix A). The four new compounds detected were designated as follows:

*Radiocoumarone A* (**1**), white amorphous powder, ${\left[\alpha \right]}_{D}^{20}$–14.4 (*c* 0.1, MeOH); UV λ_max_ MeOH nm (log ε): 215 (2.27), 275 (0.61); ESIMS *m*/*z* 231 [M + Na]^+^, HRESIMS *m*/*z* 231.0626 [M + Na]^+^ (calcd for C_11_H_12_O_4_Na, 231.0628); ^1^H and ^13^C NMR, see Table 1 (Appendix A);

*Radiocoumarone B* (**2**), white amorphous powder, ${\left[\alpha \right]}_{D}^{20}$–18.8 (*c* 0.1, MeOH); UV λ_max_ MeOH nm (log ε): 215 (2.28), 275 (0.62); ESIMS *m*/*z* 261 [M + Na]^+^, HRESIMS *m*/*z* 261.0732 [M + Na]^+^ (calcd for C_12_H_14_O_5_Na, 261.0733); ^1^H and ^13^C NMR, see Table 1 (Appendix A).

*Radiocoumarone C* (**3**), white amorphous powder, ${\left[\alpha \right]}_{D}^{20}$–72.8 (*c* 0.1, MeOH); UV λ_max_ MeOH nm (log ε): 213 (2.26), 275 (0.60); ESIMS *m*/*z* 291 [M + Na]^+^, HRESIMS *m*/*z* 291.0831 [M + Na]^+^ (calcd for C₁₃H₁₆O₆Na,291.0839); ^1^H and ^13^C NMR, see Table 1 (Appendix A).

*Radiocoumarone D* (**4**), white amorphous powder, ${\left[\alpha \right]}_{D}^{20}$–6.7 (*c* 0.1, MeOH); UV λ _max_ MeOH nm (log ε): 216 (2.27), 278 (0.62); ESIMS *m*/*z* 415 [M + H]^+^, HRESIMS *m*/*z* 415.1381 [M + H]^+^ (calcd for C_22_H_23_O_8_, 415.1387); ^1^H and ^13^C NMR, see Table 2 (Appendix A).

### 3.3. Inhibitory Effects of Mushroom Tyrosinase

The tyrosinase inhibition assay was achieved according to a previously described method [21]. Both the enzyme (tyrosinase, EC 1.14.18.1) and substrate (l-3,4-dihydroxyphenylalanine) (l-DOPA) were purchased from Sigma-Aldrich. All test compounds were first dissolved in 5% dimethyl sulfoxide (DMSO) (80 μL), and the tyrosinase inhibitory activity was evaluated at various concentrations (5 to 200 μM). The reaction mixture consisted of 40 μL of tyrosinase solution (125 unit/mL in 67 mM phosphate buffer, pH 6.8), 40 μL of L-DOPA (25 mM in 67 mM phosphate buffer, pH 6.8), and 80 μL of the test sample. After incubation at 37 °C for 30 min, the amount of released DOPA chrome was measured at 470 nm using an ELISA reader (Infinite F200, Tecan Austria GmBH, Grodig, Austria). The tyrosinase inhibition (%) was calculated, and the half-maximal inhibitory concentration (IC_50_) was evaluated using linear regression analysis of the inhibitory effects under the assay conditions. Kojic acid was applied as the positive control. All assays were carried out in triplicate. Kinetic parameters were determined using the Lineweaver-Burk plots methods at increasing concentration of substrates and inhibitor. The data was calculated using Sigma-plot (SPCC Inc., Chicago, IL, USA).

### 3.4. Statistical Analysis

Data for the in vitro analyses of tyrosinase inhibition were analyzed using the Proc GLM procedure of SAS software (version 9.3, SAS Institute Inc., Cary, NC, USA). The results are reported as the least square mean values and standard deviation. Statistical significance was considered at *p* < 0.05.

## 4. Conclusions

The present study determined that 4-methylumbelliferone is readily converted into four novel dihydrocoumarins **1**–**4**. Compared to the original 4-methylumbelliferone, the new compound **4** showed potent inhibitory effects toward mushroom tyrosinase. These results will simplify the structure–activity relationship investigations of the tyrosinase inhibitory properties of dihydrocoumarin linked with hydroxy alkyl groups at C-3 and C-4 positions as compared to parent coumarin containing an olefin unit at C-3 and C-4 positions. This study proposes the convenient hydrogenation and hydroxy alkylation of major naturally occurring secondary metabolites induced by gamma rays and provides a unique approach to the synthesis of coumarin-based structure modification to deliver compounds with highly enhanced potency for tyrosinase inhibition. A further systematic study into the influences of gamma irradiation on the semi-synthesis and biological potencies of other natural secondary metabolites is currently underway.

## Figures and Tables

**Figure 1 molecules-29-00341-f001:**
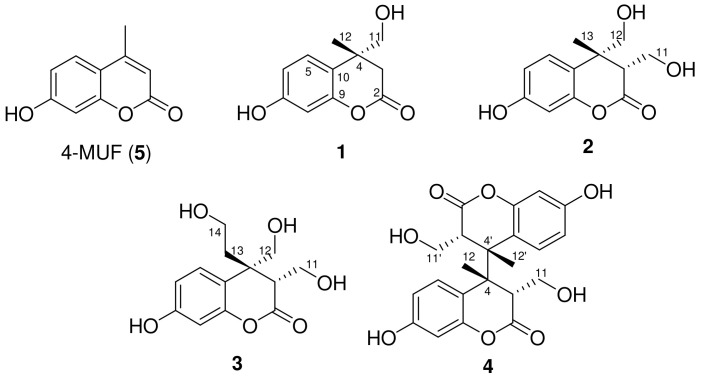
Structures of newly generated products **1**–**4** induced by gamma irradiation.

**Figure 2 molecules-29-00341-f002:**
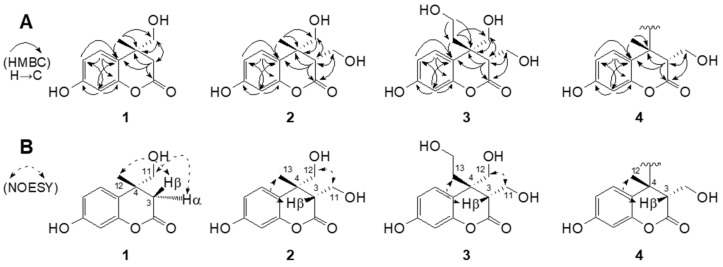
Key HMBC (**A**) and selected NOESY (**B**) correlations of **1**–**4**.

**Figure 3 molecules-29-00341-f003:**
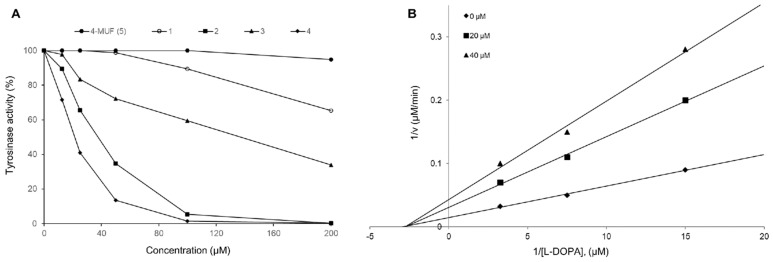
Effects of compounds **1**–**4** on the inhibition (%) of tyrosinase (**A**) and Lineweaver–Burk plots for the inhibition of tyrosinase with radiocoumarone D (**B**).

**Table 1 molecules-29-00341-t001:** ^1^H and ^13^C NMR shifts of compounds **1**–**3** in CD_3_OD ^1^.

	1		2		3	
Position	*δ*_H_ (*J* in Hz) ^2^	*δ*_C_, Type	*δ*_H_ (*J* in Hz)	*δ*_C_, Type	*δ*_H_ (*J* in Hz)	*δ*_C_, Type
2	—	178.6, C	—	180.9, C	—	180.2, C
3β	3.05 (d, 16.8)	41.5, CH_2_	2.91 (t, 4.2)	53.6, CH	3.06 (t, 4.2)	52.6, CH
3α	2.63 (d, 16.8)					
4	—	42.5, C	—	44.3, C	—	47.1, C
5	6.86 (d, 8.4)	127.0, CH	6.72 (d, 8.4)	126.9, CH	7.08 (d, 8.4)	127.3, CH
6	6.28 (dd, 8.4, 2.4)	106.1, CH	6.31 (dd, 8.4, 2.4)	106.4, CH	6.30 (dd, 8.4, 2.4)	105.8, CH
7	—	157.2, C	—	157.1, C	—	157.4, C
8	6.31 (d, 2.4)	102.7, CH	6.35 (d, 2.4)	102.8, CH	6.34 (d, 2.4)	103.3, CH
9	—	155.9, C	—	155.5, C	—	156.2, C
10	—	121.5, C	—	119.3, C	—	119.1, C
11	4.59 (d, 9.0)	79.1, CH_2_	3.79 (dd, 11.4, 4.2)	60.3, CH_2_	4.13 (dd, 11.4, 4.2)	59.0, CH_2_
	4.47 (d, 9.0)		3.66 (dd, 11.4, 4.2)		4.06 (dd, 11.4, 4.2)	
12	1.45 (s)	24.8, CH_3_	4.84 (d, 7.8)	77.9, CH_2_	4.80 (d, 9.0)	76.0, CH_2_
			4.32 (d, 7.8)		4.61 (d, 9.0)	
13			1.45 (s)	26.4, CH_3_	2.63 (m)	34.2, CH_2_
					2.10 (m)	
14					3.42 (m)	58.8, CH_2_
					3.29 (m)	

^1^ Assignments of chemical shifts are based on the analysis of HSQC and HMBC spectra. ^2^ *J* values (Hz) are given in parentheses.

**Table 2 molecules-29-00341-t002:** ^1^H and ^13^C NMR shifts of bisdihydrocoumarin **4** in acetone-*d*_6_ ^1^.

Position	*δ*_H_ (*J* in Hz) ^2^	*δ*_C_, Type	Position	*δ*_H_ (*J* in Hz)	*δ*_C_, Type
2	—	169.7, C	2′	—	169.7, C
3β	3.22 (dd, 7.2, 5.4)	48.0, CH	3β′	3.22 (dd, 7.2, 5.4)	48.0, CH
4	—	46.5, C	4′	—	46.5, C
5	6.47 (d, 8.4)	131.0, CH	5′	6.47 (d, 8.4)	131.0, CH
6	6.43 (dd, 8.4, 2.4)	110.9, CH	6′	6.43 (dd, 8.4, 2.4)	110.9, CH
7	—	158.2, C	7′	—	158.2, C
8	6.36 (d, 2.4)	102.5, CH	8′	6.36 (d, 2.4)	102.5, CH
9	—	152.2, C	9′	—	152.2, C
10	—	115.4, C	10′	—	115.4, C
11	3.85 (dd, 11.4, 5.4)	60.9, CH_2_	11′	3.85 (dd, 11.4, 5.4)	60.9, CH_2_
	3.49 (dd, 11.4, 7.2)			3.49 (dd, 11.4, 7.2)	
12	1.28 (s)	19.5, CH_3_	12′	1.28 (s)	19.5, CH_3_

^1^ Assignments of chemical shifts are based on the analysis of HSQC and HMBC spectra. ^2^ *J* values (Hz) are given in parentheses.

**Table 3 molecules-29-00341-t003:** Tyrosinase inhibitory effects of the isolated dihydrocoumarins **1**–**4**.

Compound	Tyrosinase IC_50_ Value (μM) ^1^	Inhibition Type (*Ki*, μM)
Gamma-ray treated 4-MUF (50 kGy)	86.7 ± 1.6 ^2^	NT ^3^
4-MUF (**5**)	>500 ^a^	NT
**1**	142.9 ± 1.5 ^b^	NT
**2**	49.0 ± 1.3 ^c^	NT
**3**	254.6 ± 2.1 ^a^	NT
**4**	19.8 ± 0.5 ^d^	Non-competitive (16.0 ± 0.2)
Kojic acid ^4^	131.7 ± 1.7 ^b^	NT

^1^ Tested compounds were examined in triplicate experiments. Different letters (a–d) within the same column indicate significant differences (*p* < 0.05). ^2^ Results expressed as IC_50_ value using μg/mL unit. ^3^ NT: not tested. ^4^ Kojic acid was used as a positive control.

## Data Availability

The data presented in this study are available on request from the corresponding author.

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
