# Peer review of "Novel Dihydrocoumarins Induced by Radiolysis as Potent Tyrosinase Inhibitors"

_molecules, 2024, doi:10.3390/molecules29020341_

Round 1

Reviewer 1 Report

Comments and Suggestions for Authors

In the present manuscript, Gyeong et. al. successfully synthesized four newly generated dihydrocoumarin by exposing 4-methylumbelliferone to gamma rays and characterized all the newly synthesized compounds.

1.     It would be nice to have a schematic presentation of Dopachrome formation from tyrosine for the readers to understand the mechanism. The authors may refer- J. Agric. Food Chem. 2003, 51, 10, 2837–2853.

2.     The authors should perform a toxicity study in melanoma cell line with compound 2 and 4 having increased inhibitory activities (Ref: South African Journal of Botany 74 (2008) 577–582). This will bring more impact to this study.

3.     The authors should include dose-response curve of compounds 1-4 while representing the inhibitory effect data.

4.     In fig 1, change the number of 4-MUF. It should be 5.

5.     Include reference for line 66-69.

6.     In line 190, table 2 should be table 3.

Reviewer 2 Report

Comments and Suggestions for Authors

This manuscript reported the application of gamma-irradiation on the semi-synthesis of dihydrocoumarins with increased tyrosinase inhibitory activities. As a result, four undescribed products from 4-methylumbelliferone were yielded. Their structures were established by interpretation of spectroscopic data. This work was important and worth publishing in this journal.

However, there are some concerns as following, which requires major revision.

1. It was said that very few studies have researched the influence of gamma radiolysis on the phenolic constituents widely distributed in natural sources. Please provide some examples in the Introduction.

2. It is important to give the text description of the key HMBC correlations for new compounds 14 apart from Figure 2, such as those to confirm the linkage points of the methyl and hydroxymethyl functionalities in compound 1, or the connective positions of one methyl and two hydroxymethyl units in compound 2.

3. Because the hydroxymethyl is different from the ethyl, it is not so reliable to determine the absolute configuration of compound 1 by comparison the sign of specific optical rotation value of 4-ethyl-4-methylchroman-2-one. Indeed, both compound 1 and 4-ethyl-4-methylchroman-2-one shared the same chromophore with the chiral center, it is possible to assign the absolute configuration by comparison of their CD spectra. Additionally, other methods such as QM-NMR and ECD calculation, could be applied. Please assign the absolute configurations of other three new compounds.

4. The numbering of compound 2 in Figure 1 was different from that shown in Figure 2.

5. It is difficult to understand the expression ‘the structure of compound 3 was inferred as radiocoumarone C, a new degraded product of 3’.

Others:

1. It is better to use only ‘gamma’ or ‘γ’, not both of them, in the manuscript.

2. P4L124: ‘…one less hydroxymethyl residue than that of 1’ → ‘…one more hydroxymethyl residue than that of 1

3. P8L281: ‘alky’ → ‘alkyl’

Comments on the Quality of English Language

There are few grammar or typo errors, which are given in the comments to the authors.

Round 2

Reviewer 2 Report

Comments and Suggestions for Authors

The authors nicely addressed the majority of my comments and revised accordingly. However, revisions were still requested.

1.      P1L26: ‘the potent metabolites 4 have non-competitive modes of action’ → ‘the potent metabolite 4 had non-competitive mode of action’

2.     The numberings of 11 and 12 in compound 2 shown in Figure 2 were wrong. Please revise them according to those of Figure 1.

3.     P3L123: ‘the location of’ → ‘the locations of’

4.     P3L125: ‘were characterized’ → ‘was characterized’

5.     P4L136: ‘The connective positions of one methyl and two hydroxymethyl units in 2 was…’ → ‘The connective positions of one methyl and two hydroxymethyl units in 2 were…’

6.     P4L138: ‘…and H-12 (δH 4.84 and 4.32)/H-13 (δH 1.45)to…’ → ‘…and from H-12 (δH 4.84 and 4.32)/H-13 (δH 1.45) to…’

7.     P5L159: ‘The a further partial structure…’ → ‘The further partial structure…

8.     P5L161: ‘HMBC correlation’ → ‘HMBC correlations

9.     P5L162: ‘the location of the hydroxymethyl groups’ → ‘the locations of the hydroxymethyl groups

10.  P5L163: ‘The relative configuration’ → ‘The relative configurations

11.  P5L164: ‘the NOESY spectrum’ → ‘the NOESY correlations

12.  P6L211: ‘The Ki values’ → ‘The Ki value

13.  P6L212: ‘thant’ → ‘that

14.  P6L213: ‘ring, and were found’ → ‘ring was found

Comments on the Quality of English Language

There are some grammar or typo errors, which are given in the comments to the authors.
